# Anticoagulation in Patients with End-Stage Renal Disease: A Critical Review

**DOI:** 10.3390/healthcare13121373

**Published:** 2025-06-08

**Authors:** FNU Parul, Tanya Ratnani, Sachin Subramani, Hitesh Bhatia, Rehab Emad Ashmawy, Nandini Nair, Kshitij Manchanda, Onyekachi Emmanuel Anyagwa, Nirja Kaka, Neil Patel, Yashendra Sethi, Anusha Kavarthapu, Inderbir Padda

**Affiliations:** 1PearResearch, Dehradun 248001, India; fnuparul@gmail.com (F.P.); ratnanitanya98@gmail.com (T.R.); sachinsachu930@gmail.com (S.S.); hitesh.bhatia@guthrie.org (H.B.); rehab.sayed@med.helwan.edu.eg (R.E.A.); dr19nandini@gmail.com (N.N.); kshitijmanchanda123@gmail.com (K.M.); emmanuelonyeka96@gmail.com (O.E.A.); nirja@pearresearch.com (N.K.); neil@pearresearch.com (N.P.); 2Department of Medicine, University of Michigan Health-Sparrow Hospital, Michigan State University, Lansing, MI 48912, USA; 3Department of Medicine, Chhattisgarh Institute of Medical Sciences, Bilaspur 495001, India; 4Department of Medicine, ESIC Medical College, Faridabad 121001, India; 5Department of Medicine, Guthrie Robert Packer Hospital, Sayre, PA 18840, USA; 6Department of Medicine, Helwan University, Helwan 11711, Egypt; 7Department of Medicine, Byramjee Jeejeebhoy Medical College, Pune 411001, India; 8Department of Medicine, Government Medical College, Surat 395001, India; 9Department of Medicine, New Vision University, Tbilisi 0159, Georgia; 10Department of Medicine, GMERS Medical College, Himmatnagar 383001, India; 11Department of Medicine, Subharti Medical College, Meerut 250005, India; 12Department of Medicine, Richmond University Medical Center, SUNY Downstate Health Sciences University, Brooklyn, NY 11203, USA; akavarthapu@rumcsi.org

**Keywords:** anticoagulation, CKD, ESRD, DOAC, warfarin, heparin

## Abstract

Background: Chronic kidney disease (CKD) and its advanced stage, end-stage renal disease (ESRD), affect millions worldwide and are associated with a paradoxical hemostatic imbalance—marked by both increased thrombotic and bleeding risks—which complicates anticoagulant use and demands clearer, evidence-based clinical guidance. Design: This study is a critical review synthesizing the current literature on anticoagulant therapy in CKD and ESRD, with emphasis on altered pharmacokinetics, clinical complications, and therapeutic adjustments. Data Sources: PubMed, Scopus, and Google Scholar were searched for articles discussing anticoagulation in CKD/ESRD, focusing on pharmacokinetics, clinical outcomes, and dosing recommendations. Study Selection: Studies examining the safety, efficacy, and pharmacokinetics of anticoagulants—including heparin, low-molecular-weight heparin (LMWH), warfarin, and direct oral anticoagulants (DOACs)—in CKD and ESRD populations were included. Data Extraction and Synthesis: Key findings were summarized to highlight the dose modifications, therapeutic considerations, and clinical challenges in managing anticoagulation in CKD/patients with ESRD. Emphasis was placed on balancing thrombotic and bleeding risks and identifying gaps in existing guidelines. Results: Patients with CKD and ESRD exhibit a paradoxical hypercoagulable state marked by platelet dysfunction, altered coagulation factors, and vascular endothelial damage. This condition increases the risk of thrombotic events, such as deep vein thrombosis (DVT) and pulmonary embolism (PE), while simultaneously elevating bleeding risks. Hemodialysis and CKD-associated variables further complicate the management of coagulation. Among anticoagulants, unfractionated heparin (UFH) is preferred due to its short half-life and adjustability based on activated partial thromboplastin time (aPTT). Low-molecular-weight heparins (LMWHs) offer predictable pharmacokinetics but require dose adjustments in CKD stages 4 and 5 due to reduced clearance. Warfarin necessitates careful dosing based on the estimated glomerular filtration rate (eGFR) to maintain an international normalized ratio (INR) ≤ 4, minimizing bleeding risks. Direct oral anticoagulants (DOACs), particularly Apixaban, are recommended for patients with eGFR < 15 mL/min or those on dialysis, although data on other DOACs in CKD remain limited. The lack of comprehensive guidelines for anticoagulant use in CKD and ESRD highlights the need for individualized, patient-centered approaches that account for comorbidities, genetics, and clinical context. Conclusions: Managing anticoagulation in CKD/ESRD is challenging due to complex coagulation profiles and altered pharmacokinetics. Judicious dosing, close monitoring, and patient-centered care are critical. High-quality randomized controlled trials are needed to establish clear guidelines and optimize therapy for this vulnerable population.

## 1. Introduction

Chronic kidney disease (CKD) is defined by the Kidney Disease Improving Global Outcomes (KDIGO) as “abnormalities of kidney structure or function, present for ≥3 months, with implications for health” [1]. It is a common disorder affecting more than 800 million individuals worldwide—more than 10% of the world’s population [2]. End-stage renal disease (ESRD) is the most advanced stage of CKD. It is defined as the stage of kidney disease when the glomerular filtration rate (GFR) falls below 15 mL/min/1.73 m^2^ or when the patient requires dialysis (KDIGO) [1]. The global burden of ESRD is tremendous, with approximately 3.9 million patients currently receiving renal replacement therapy (RRT) such as dialysis or renal transplantation, and the numbers are expected to rise between 5 and 10 million by 2030 [3,4]. Patients with ESRD have a significantly higher risk of morbidity and mortality compared to the general population, with cardiovascular disease and sepsis/infection being the leading causes of death [5,6].

One of the major cardiovascular consequences of ESRD is hypercoagulability. Multiple factors contribute to a hypercoagulable state in CKD/patients with ESRD, including platelet dysfunction, higher levels of procoagulant factors such as factors VII, VIII, IX, XI, VII, fibrinogen, and von Willebrand factor, and low levels of natural anticoagulant factors such as protein C, protein S, and antithrombin III. In addition, vascular endothelial damage by inflammatory mediators and oxidative stress further increases the risk of thrombosis in these patients. Furthermore, hemodialysis (HD) used in kidney failure also aggravates this prothrombotic state by the activation of platelets and the coagulation cascade. The use of antiplatelet agents, such as aspirin and clopidogrel, in CKD/ESRD for coexisting vascular disease further affects the situation severely [7,8,9]. Patients with CKD face a heightened risk of cardiovascular comorbidities, including venous thromboembolism (VTE), which encompasses deep vein thrombosis (DVT) and pulmonary embolism (PE), often necessitating anticoagulation (AC) [10]. In stage 3 or 4 CKD, the adjusted relative risk of DVT is 1.71 (95% CI, 1.18–2.49) compared to individuals with normal kidney function [11]. Among patients with end-stage kidney disease (ESKD) on dialysis, the incidence of PE is over twice as high as in those with normal kidney function [12]. CKD also significantly increases the risk of atrial fibrillation (AF), with prevalence rising alongside disease progression; 10–25% of dialysis-dependent patients with CKD exhibit AF [13,14,15,16].

Patients with renal insufficiency face a dual challenge of increased susceptibility to thrombotic events and bleeding complications, representing two extremes of a complex hemostatic spectrum. On one hand, conditions such as VT and PE arise due to a hypercoagulable state. On the other hand, impaired platelet aggregation, uremic toxins, and reduced production of coagulation factors contribute to an elevated bleeding risk [9]. Other complications of thrombosis in patients with ESRD include HD access site thrombosis, arterio-venous malformations, hematoma formation, catheter-related infections, sepsis, and ischemic stroke. These serious complications can lead to significant morbidity and mortality in this population.

The hypercoagulable state in SRD presents as a paradox, forcing a clinical challenge in day-to-day practice. Despite various updates, the currently available guidelines seem elusive and do not cover the spectrum of diverse patients, leading to a lack of personalized therapy and poor outcomes. The use of anticoagulation in this population is challenging due to a substantial increased risk of bleeding, particularly in the setting of RRT. This critical review aims to appraise and compile the current body of evidence on the use of anticoagulants in patients with CKD especially ESRD. Given the complex pharmacokinetics of anticoagulants in renal impairment and the lack of unified, evidence-based guidelines, this review aims to clarify therapeutic considerations and bridge clinical knowledge gaps. By consolidating current data and highlighting key clinical insights, this review also proposes a simplified, practical algorithm to support clinicians in making safer, more individualized anticoagulation decisions for patients with ESRD.

## 2. Methodology

### 2.1. Search Strategy and Data Sources

This critical review synthesized research from PubMed, Scopus, and Google Scholar, focusing on studies published up to 2024. Search terms included combinations of “anticoagulation”, “chronic kidney disease (CKD)”, “end-stage renal disease (ESRD)”, “heparin”, “low molecular weight heparin (LMWH)”, “warfarin”, “direct oral anticoagulants (DOACs)”, “apixaban”, “rivaroxaban”, “dabigatran”, “edoxaban”, “pharmacokinetics in CKD”, “bleeding risk in renal impairment”, “thromboembolic events”, “venous thromboembolism (VTE)”, “atrial fibrillation in CKD”, “dialysis-related thrombosis”, and “KDIGO anticoagulation guidelines.” The articles were limited to English-language publications, emphasizing clinical studies, systematic reviews, meta-analyses, and guideline-based evidence.

### 2.2. Study Selection

Studies were included if they assessed the safety, efficacy, or pharmacokinetics of anticoagulants in populations with chronic kidney disease (CKD) or end-stage renal disease (ESRD). Eligible studies focused on therapeutic strategies, dose modifications, and clinical outcomes related to unfractionated heparin (UFH), low-molecular-weight heparin (LMWH), warfarin, and direct oral anticoagulants (DOACs). Additionally, articles addressing the 2022 Kidney Disease: Improving Global Outcomes (KDIGO) guidelines and their updates on anticoagulation practices were reviewed. Studies that did not pertain to CKD/ESRD or anticoagulation, or lacked sufficient clinical relevance, were excluded from the analysis.

### 2.3. Data Extraction and Synthesis

Relevant data, including study objectives, population characteristics, anticoagulant pharmacokinetics, dosing modifications, clinical outcomes, and associated challenges, were extracted. Key findings were categorized into thematic areas: (1) pharmacokinetics and pharmacodynamics of anticoagulants in CKD/ESRD, (2) thrombotic and bleeding complications, (3) dosing recommendations for different anticoagulants, and (4) gaps in clinical guidelines. Updates from the recent KDIGO guidelines were integrated to provide a comprehensive view of evolving best practices and recommendations. The data were critically analyzed to identify trends, therapeutic strategies, and unmet needs in anticoagulation management for CKD/patients with ESRD.

### 2.4. Analysis Framework

This review adopted a narrative synthesis approach to integrate findings from diverse studies. The focus was on highlighting the interplay between thrombotic risks and bleeding risks in CKD/ESRD, with specific attention to anticoagulant-specific considerations such as renal clearance, monitoring requirements, and dosing precision. Additionally, gaps in existing guidelines and updates from the KDIGO recommendations were examined to emphasize patient-centered approaches and future research priorities.

### 2.5. Ethical Considerations

This review utilized publicly available data and did not involve human subjects or require institutional ethical approval.

## 3. The Hypercoagulability Paradox in CKD

Cardiovascular causes contribute to 48% of all deaths in patients with CKD. These include atrial fibrillation, thromboembolism, and bleeding [17]. Patients with CKD and ESRD are at an increased risk of both bleeding as well as thrombosis due to platelet dysfunction stemming from deranged vWF and platelet interactions. The increased prothrombotic events precipitate as pulmonary embolism, or arteriovenous fistula thrombosis, acute coronary syndrome, transient ischemic attack, and deep venous thrombosis [18]. Patients with CKD on peritoneal dialysis have increased coagulation factors due to the activation of macrophages and thromboplastin as the dialysate is exposed to the peritoneum. HD patients reportedly have lower fibrinogen levels, contributing to an increased risk of thrombosis [19]. However, other studies have shown the effect of reduced vWf and increased tissue-type plasminogen activator (tPA) levels noted in patients after hemodialysis, which is associated with higher bleeding risk in such patients. Intrinsic platelet defects combined with abnormal platelet vessel wall interactions and altered calcium levels in patients with ESRD also contribute to increasing bleeding diathesis [20]. RRT can only partially treat these defects and not completely eliminate them [21]. Various factors contribute to the paradox, as detailed below.

### 3.1. Factors Increasing the Bleeding Risk

In platelets, the ATP:ADP ratio is increased, and serotonin content is decreased, which further alters the balance between the pro- and anti-coagulation factors. The platelet defect is multifactorial ranging from impaired arachidonic acid release from platelets to storage pool defect, and while the former is improved with RRT, the latter is not [22]. Additionally, the NO levels increase in uremic patients, inhibiting platelet aggregation [23].

Erythrocytes play a role in stimulating platelet ADP release and inactivating prostacyclin (PGI2), which further scavenges NO and contributes to bleeding risk [24]. Antiplatelets and anticoagulants administered to decrease the thrombotic risk or during hemodialysis contribute to bleeding [20]. The platelet surface receptors GpIb and GpIIbIIIa are proteolyzed more in uremic patients, thus decreasing the clot formation potential [25].

The bleeding risk might also be increased in the initial stages of CKD, especially if associated with albuminuria. The association is more potent with albuminuria than eGFR levels initially but not with worsened eGFR levels, where the bleeding risk increases proportionately with decreasing GFR. This finding reflects the shared mechanism of underlying dysfunctional endothelium for both bleeding and albuminuria [26].

### 3.2. Factors Increasing the Thrombotic Risk

The increased levels of several clotting factors, such as XII, VII, and thrombin, along with decreased antithrombin activity, contribute to thrombosis. The activity of tPA is decreased due to elevated plasminogen activator inhibitor-1 (PAI-1) and homocysteine, thus preventing clot breakdown [27,28]. Proinflammatory markers, such as CRP and IL-6, also contribute to clot formation, in addition to increased fibrinogen and tissue factors [29].

The expression of phosphatidylserine is also increased on the surfaces of platelets, which binds factors V and X, leading to prothrombin activation and thrombus formation [30]. The loss of antithrombotic properties also plays a major role [27]. These patients also tend to be in a catabolic state leading to increased muscle breakdown, further contributing to decreased NO levels [31]. There is also some evidence that supports decreased anti-phospholipid antibodies in HD patients may contribute to endothelial and platelet dysfunction [32]. The role of homocysteine is increasingly studied in thrombus formation. Its role in fibrin generation and matrix metalloproteinase–9 activation by gene induction contributes to its role in increasing thrombosis [27].

Research on microparticles (MPs) in kidney diseases has shown promising results over recent decades. Studies on both plasma and urinary MPs have revealed their potential as biomarkers for early diagnosis and prognosis in diabetic kidney disease (DKD). Alterations in MP number and composition have been documented in both Type 1 and Type 2 diabetes mellitus, suggesting their utility in early DKD detection and monitoring [33,34]. Additionally, dialysis treatment has been found to trigger MP release, with notable differences observed between high-flux and low-flux dialyzers [35]. Microparticles (MPs) released from various cells like endothelial cells, platelets, and monocytes/macrophages during cell aging, differentiation, and trauma have a role in both bleeding and thrombus formation [33]. MPs have phosphatidylserine (PS) and tissue factor (TF) on their surface. MPs also have miRNAs. PS and TF facilitate thrombin formation through the coagulation cascade. miRNAs are involved in regulating P2Y12 receptor expression, thus resulting in increased platelet aggregation. They are also found in increased levels in less active platelets [20]. MPs’ role as an anticoagulant is by its tissue factor pathway inhibitor (TFPI) that is expressed on its surface. Also, MPs have thrombomodulin (TM) and Endothelial cell protein C receptor (EPCR), which inactivates factors Va and VIIIa via protein C [33]. While these findings are promising, further research is needed to fully establish the clinical impact and practical applications of MP analysis in nephrology.

In conclusion, a hypercoagulability paradox is noted in End-stage renal disease patients on hemodialysis, which consists of a combination of coagulopathy and increased bleeding risk manifesting because of factors like increased final clot strength, reduced clot breakdown and platelet dysfunction, respectively [19].

## 4. The Severity of CKD-ESRD and Coagulation Abnormalities

Chronic renal disease is associated with many hemostatic abnormalities due to multiple reasons like defective platelet aggregation, reduced platelet adhesiveness, reduced platelet factor 3 availability, etc. Studies have shown that plasma metabolites like urea, phenolic acid, guanidinosuccinic acid have a role in inhibiting the function of platelets. Other associated factors like defective binding of platelets to the subendothelium, and abnormal platelet arachidonic acid metabolism have a role in contributing to the hemostatic problem in severe end-stage renal disease patients. Clinically, uremic bleeding disorder manifests as increased bleeding time. HD is the mainstay of treatment and prevention of uremic bleeding, but it is reasonable to use DDAVP and cryoprecipitate for temporary stabilization, especially before any invasive procedures [36].

The severity of chronic kidney disease can be stratified into five stages (Table 1).

Several factors affect the severity and progression of CKD into ESRD. Many studies have been undertaken to demonstrate the effect of pre-existing kidney disease on the progression of chronic kidney disease to end-stage renal disease. Quantitative parameters like estimated glomerular filtration rate and proteinuria were assessed to compare various groups of CKD like adult polycystic kidney disease, IgA nephropathy, diabetic nephropathy, and hypertensive nephropathy. On analysis, it was noted that the progression of CKD in patients of ADPKD and IgA nephropathy was slower than in the other groups and the addition of renin–angiotensin–aldosterone blockers had significantly prolonged the period between the first visit to the nephrologist and the first renal replacement treatment. Other factors like dietary protein intake and blood pressure control in renal disease patients have been studied in the Modification of Diet in Renal Disease Study (MDRD) to determine their influence on the progression and severity of chronic renal disease as well. Maintaining lower blood pressure has been helpful in slowing down the disease course by roughly 29% and reducing the intake of dietary protein was beneficial in patients with an estimated GFR of less than 25 mL/min [37,38].

Other demographic factors like race have demonstrated considerable variation in the severity and manifestation of CKD among the population. For instance, African American (AA) populations progress to ESRD more often as compared to white American populations. This could be attributed to the increased prevalence of uncontrolled hypertension and diabetes mellitus among African American populations, making them more predisposed to developing CKD in the first place. Racial differences in the rates of tubular clearance of creatinine, physiological differences in the body mass index, and metabolism are reflected in the higher level of serum creatinine seen among African American populations in the United States. This was measured using gold-standard iothalamate clearance to study a large cohort of the AA population. This difference is attributed to the increased risk of progression compared to white populations. Other factors like socioeconomic variability and lifestyle also significantly contribute to the severity of CKD in a given population. Genetic influence of genes like plasma kallikrein and human homolog of the rodent renal failure 1 (RRF-1) gene have been associated with the prevalence of ESRD among AA populations [39].

Studies to emphasize the presence of Fibroblast Growth Factor-23 (FGF-23) as a risk factor for mortality and higher morbidity in CKD have been successful. A higher level of FGF-23 is more strongly associated with ESRD than other CKD risk factors. FGF-23 is a biomarker of elevated parathyroid hormone and serum phosphate, which corresponds to impaired kidney function. The importance of FGF-23 levels in the progression of CKD through various stages like stage 2 through stage 4 has been evaluated in recent times. Thus, recent studies have established raised FGF-23 levels as a predictor of adverse outcomes in patients with CKD [40].

## 5. Hypercoagulability and Calciphylaxis: The Double Trouble

Calciphylaxis is a rare but life-threatening vasculopathy that results from the deposition of calcium in microvessels in subcutaneous adipose tissue and the deeper dermis. It results in ischemic lesions, which cause difficulty in managing pain. Calciphylaxis associated with advanced CKD and ESRD is also called calcemic uremic arteriopathy (CUA) [41].

In this population, the release of 1-alpha-hydroxylase is compromised, which leads to a decreased synthesis of vitamin D. Vitamin D has a prominent role in regulating calcium and phosphorus reabsorption and excretion in the renal tubular cells. In the setting of impaired kidney function, altered calcium reabsorption, and compromised phosphorus excretion can result in secondary hyperparathyroidism, which, in turn, promotes bone resorption, increasing serum calcium levels and enhancing the risk of microvascular calcifications and renal osteodystrophy [42].

Some other rare non-uremic causes of calciphylaxis include obesity, diabetes mellitus, female sex, hyperparathyroidism, warfarin, corticosteroids, vitamin K deficiency, vitamin D deficiency, hypoalbuminemia, protein C and S deficiencies, Crohn’s disease, autoimmune disorders, substantial weight loss, recurrent hypotension, and malignant neoplasms (cholangiocarcinoma, hematologic malignancies, and melanoma) [43]. The nonuremic causes are less commonly associated with mortality as compared to the uremic causes. Although the data on incidence of calciphylaxis remain limited, the largest nationwide study to date estimates an incidence rate of 3.49 per 1000 patient years among patients on HD [44]. Further, one-year mortality in patients diagnosed with calciphylaxis is 45–80% [45].

Microvascular calcification is integral to the pathogenesis of calciphylaxis, but other factors including recurrent vascular endothelial damage and abnormal adipocyte proinflammatory cytokine signaling also play an important role [46,47].

It is well established that the hypercoagulable state plays an important role in the development of calciphylaxis. The presence of lupus anticoagulant, protein C deficiency, and mixed thrombophilias has been substantially linked with calciphylaxis [48,49]. The presence of fibrin thrombi in skin biopsies has been associated with the degree of clinical pain; therefore, the development of microvascular thrombosis may help to explain the characteristic acute soreness of calciphylaxis [50]. The pathophysiology of calciphylaxis has been linked to abnormalities in the molecular mediators that ordinarily prevent microvascular calcification. The extracellular matrix peptide carboxylated matrix G1a protein (MGP), produced by vascular and endothelial cells, is an anti-calcifying protein found in the extracellular matrix. The vitamin K epoxide reductase inhibitor warfarin, which is frequently prescribed to dialysis patients, blocks the gamma-carboxylation necessary for the activity of MGP. Warfarin thus intensifies the vascular ossification–calcification processes, which may account for the substantial correlation between warfarin usage and the emergence of CUA. The expression of the additional calcification inhibitors fetuin-A and pyrophosphate is also reduced in CKD and calciphylaxis [43].

Ultimately, low levels of MGP, fetuin-A, and pyrophosphate result in a favorable environment for calcification to occur. The hallmark presentation of cutaneous calciphylaxis is excruciating painful necrotic ulcerations often occurring with other manifestations like livedo racemosa or livedo reticularis (net of lace-like appearance on the limbs and rarely on the trunk), ulcerated patches or plaques, hemorrhagic bullae or necrotic eschars [41,47].

## 6. Diagnostic Markers and Their Accuracy with Disease Severity

Chronic renal insufficiency can have a significant impact on hemostasis, through a variety of processes, leading to either a procoagulant state with recurrent blood clots or an anticoagulant state with episodes of bleeding. The risk is associated with the severity of renal dysfunction, with advanced renal failure observed to be associated with bleeding and moderately preserved renal function more associated with thrombosis [20].

Normal hemostasis can be negatively impacted by platelet, coagulation, or fibrinolytic dysfunction, posing a serious risk of bleeding and hypercoagulation, caused by alterations in coagulation regulation and increased platelet hyperreactivity. In a patient with renal insufficiency, the uremic toxins circulating in the blood partially contribute to platelet dysfunction, resulting in the release of active molecules, enhancing immune complex deposition, and altering glomerular permeability. Dialysis can improve platelet abnormalities but does not eliminate the risk of hemorrhage [51].

In a study by Huang M-J et al., 95 patients with CKD and 20 healthy controls were evaluated to study the incidence of thromboembolic incidents and their impact on the prognosis of patients with CKD using coagulation factors, anticoagulation system, conventional blood tests, standard coagulation tests, and thromboelastography (TEG). It emerged that individuals with CKD had higher levels of procoagulant factors like vWF (von Willebrand factor), FVIII (Factor VIII), D-dimer, fibrinogen, and FVII (Factor VII), and the measurement of these parameters was proportionate to the severity of the renal injury. In addition to the usual causes of thromboembolic events (such as hypertension, diabetes, obesity, and dyslipidemia), renal dysfunction itself appears to have a role in the enhanced procoagulant factors in patients with CKD [52].

## 7. The Current Management Protocols

Parenteral anticoagulants, particularly low-molecular-weight heparin (LMWH), require caution in patients with CKD due to renal clearance dependency. Direct oral anticoagulants (DOACs) pose additional challenges, especially for advanced patients with CKD (stages 4/5 and on hemodialysis), given limited safety data. Accurate dosing is critical, as many anticoagulants rely on renal elimination, making clearance rates a key consideration. For patients with AF at elevated risk for stroke and CKD stages 3/4, treatment with warfarin or, preferably, evidence-based doses of direct thrombin or factor Xa inhibitors is recommended to reduce the risk of stroke. Guidelines from the American Heart Association, American College of Cardiology, and Heart Rhythm Society (AHA/ACC/HRS) recommend warfarin across all CKD stages, while also supporting DOACs, particularly Apixaban and Rivaroxaban. However, the European Society of Cardiology only endorses Apixaban for CrCL < 15 mL/min, differing from the American and Canadian guidelines. For patients with AF at elevated risk for stroke and who have end-stage CKD (CrCl < 15 mL/min) or are on dialysis, it might be reasonable to prescribe warfarin (INR 2.0–3.0) or an evidence-based dose of Apixaban for oral anticoagulation to reduce the risk of stroke. Detailed recommendations for acute and maintenance anticoagulation use are provided in subsequent sections [53,54,55].

Warfarin use in CKD can lead to supratherapeutic international normalized ratios (INRs) due to individual variability. Dose adjustments are advised for eGFR 30–59 mL/min/1.73 m^2^ (10% reduction) and eGFR < 30 mL/min/1.73 m^2^ (19% reduction). Patients are at risk for warfarin-induced nephropathy, marked by a serum creatinine rise of ≥0.3 mg/dL within 7 days and INR > 3.0, and acute renal failure. DOACs offer advantages but require dose adjustments in renal impairment due to variable renal clearance [56]. Additionally, DOAC-induced nephropathy, caused by tubular obstruction from blood clots, is a potential complication.

### 7.1. Therapeutic Status of Parenteral Anticoagulants and Best Practice for Clinicians

Currently, available options for parenteral anticoagulation in patients with CKD are unfractionated heparin (UFH) and low-molecular-weight heparins (LMWHs), including fondaparinux. UFH is used as per its usual indications in patients with CKD. UFH, with its shorter half-life and short anticoagulant effect, is the preferred choice in patients with severe renal insufficiency who are at high risk of bleeding, providing a reassuring safety profile [57,58]. However, dosage adjustments are required in patients with advanced CKD. The initial dose should be reduced by 33%, and then the dosage should be adjusted based on aPTT [56].

LMWHs are often preferred due to their predictable pharmacokinetics and easier dosing compared to UFH. They also have a better safety profile as they have reduced interactions with platelets and endothelium, a lower risk of heparin-induced thrombocytopenia (HIT), and a reduced chance of osteopenia. Although the monitoring requirement is less than that of UFH, it might be essential to monitor in advanced CKD. Also, there is expert consensus on reduced doses versus avoiding it in CKD stages IV and V patients. In these cases, anti-factor Xa level should be ideally used to monitor the therapeutic levels, which are seldom available; aPTT is a close estimate in clinical practice [56,57].

A daily dosing regime of LMWH is preferred in patients with bleeding tendencies over the traditional twice-daily dosing. This is because there is insufficient evidence that the potential benefits of patient convenience in once-daily dosing outweigh the risks, even in patients with severe renal impairment. The initial dose should be reduced and then adjusted with anti-factor Xa levels (Figure 1). The reduced dose enoxaparin is the most widely researched LMWH regimen in patients with renal disease—1 mg/kg every 12 h is used in patients with more than 30 mL/min of creatinine clearance, while 1 mg/kg daily with dose adjustment is used in patients with less than 30 mL/min (Figure 1) [25,56,59]. Table 2 compares effectiveness and safety of antithrombotic therapy in patients with atrial fibrillation and end-stage renal disease (ESRD).

### 7.2. Therapeutic Status of Oral Anticoagulants

The use of Vitamin K antagonists in the patients of CKD is extensive even today. In this class, warfarin stays the most widely studied drug with clearly described pharmacokinetics and pharmacodynamics. Warfarin is metabolized by cytochrome P450 liver enzymes namely CYP2C9, CYP1A2, CYP2C19, and CYP3A4 with 0% renal excretion or very minimal amount excreted unchanged in urine [53]. Guidelines dictating its use in the patients with CKD vary significantly. The American Heart Association/American College of cardiology recommend warfarin in all stages of CKD with varying levels of evidence whereas Canadian guidelines recommend it in stage 4 and European guidelines do not provide any specific information. Warfarin has a narrow therapeutic window and extensive drug interactions, so it should be cautiously used in CKD. Various studies have shown that at non-adjusted doses patients on warfarin experienced a higher number of bleeding incidences within 90 days of treatment initiation [67]. For this reason, to prevent the risk of hemorrhage an average reduction in warfarin dose by 10% in patients with eGFR between 30 and 59 mL/min/1.73 m^2^ and by 19% in those with eGFR  <  30 mL/min/1.73 m^2^, is required in order to maintain INR  ≤  4 [68].

Furthermore, calciphylaxis seen with Vitamin K antagonists can lead to renal vasculature’s calcification further deteriorating renal function, pronouncing the bleeding risk with warfarin even more. Additionally, calciphylaxis leads to increased risk of infections in an already vulnerable population [56,67]

Another point of significance is the warfarin induced nephropathy which is acute renal failure at an INR threshold > 3 and carries a mortality rate of 31% at 1 year [67].

In patients on RRT especially, warfarin seems to be a better treatment of choice as it is metabolised by the liver and its plasma protein binding protects it from the removal by dialysis thereby maintaining the drug levels despite regular dialysis [69]. Furthermore, there is a paucity of evidence on the use of DOACs in patients with dialysis which again makes warfarin a better calculated choice for most clinicians. Despite all this, there have been studies that have shown a higher bleeding rate in dialysis patients started on warfarin [70] but other studies have suggested that even in patients who are not on warfarin the bleeding rate remains high because of a variety of factors related to the disease pathology itself. Ref. [71] Given that there is lack of clear guidelines on the use of warfarin in patients on dialysis and a general apprehension to use anticoagulation by clinicians in the population, which is already at an increased risk of bleeding, the use of warfarin is majorly left to the clinicians’ discretion tailored to individual patient needs with cautious consideration of risk and benefits of using it. Table 2 summarizes the findings from various studies comparing stroke and bleeding outcomes in patients with AF and ESRD receiving different antithrombotic therapies, including warfarin and DOACs.

To guide treatment better in patients CKD, it is of the great importance that high quality randomized control trials be conducted in this area so that clear guidelines can be formulated, and standardization of treatment can be carried out.

### 7.3. Therapeutic Status of Direct Oral Anticoagulants

The use of direct oral anticoagulants needs definite dose adjustments in the patients of CKD given that their elimination is primarily through kidneys. Based on the currently available studies, the most recent 2023 AF guidelines recommend use of labeled doses of DOACs in patients with stage 4 CKD in addition to warfarin (Class 2a recommendation) and preferably Direct thrombin/factor Xa inhibitors in patients with CKD stage 3 (class 1 recommendation). Further, in patients with CKD stage 5 or on HD, 2023 guidelines recommend use of warfarin or evidence-based doses of Apixaban (5 mg or 2.5 mg twice-daily (BID)) across all four CrCL-based classes. The 2.5 mg BID dose is recommended only if at least two of the following criteria are met: serum creatinine ≥ 1.5 mg/dL, age ≥ 80 years, or body weight ≤ 60 kg) (1). There are very little data to recommend the use of other DOACs in patients with advanced CKD, including stages 4/5 or on HD. Per the 2023 guidelines, Rivaroxaban can also be used in patients with CrCL < 15 at reduced dose of 15 mg once daily; however, pharmacokinetic data are limited [53].

In a previously conducted meta-analysis which compared DOACs with warfarin, it was found that DOACs when given in patients with CKD are associated with a 22% lower risk of stroke, SE or VTE than the warfarin group and a 17% less risk of major bleeding. Especially factor XA inhibitors showed better efficacy and lower bleeding risk amongst DOACs. Stage-wise comparison in patients with CKD showed a better effect by DOAC as compared to warfarin in stage 3 but similar in stages 4–5 or patients on hemodialysis [72]. Two other studies that compared Apixaban with warfarin in patients of CKD (all stages) showed that bleeding events were significantly lower in patients on Apixaban [73,74]

Further, DOACs do not pose the risk of vascular calcification warfarin does, making them a better choice in patients with CKD.

With pivotal clinical trials largely excluding the patients with CrCl < 25 mL/min, the use of DOACs should be carried out cautiously. Another study carried out on patients with CKD stages 4–5 successfully showed a reduction in all-cause mortality and overall bleeding risk with no significant difference in embolic episodes in patients of DOACs as compared to warfarin and therefore suggested DOACs as an acceptable alternative to warfarin for stroke prevention for patients of CKD with concomitant AF [75]. Amongst the HD patients with concomitant AF/VTE, two studies have shown decreased bleeding risk with Apixaban as compared to warfarin, with no significant difference in stroke/embolic events [64,76]. Given the paucity of evidence, it is recommended that patients on DOACs should be closely monitored throughout, and further high-quality randomized controlled trials targeting this population should be carried out to establish the efficacy and safety of DOACs as the anticoagulation form for patients with CKD.

## 8. Anticoagulation Trials and Special Population of Patients with ESRD

The trials on anticoagulation have mostly excluded the patients with CKD/ESRD. The ARISTOTLE study, which compared Apixaban and warfarin in patients with AF, found Apixaban to be superior to warfarin in preventing stroke or systemic embolism and caused less major bleeding and mortality, but this landmark trial excluded patients with ESRD [77]. In a retrospective cohort study carried out in 2018 by Siontis et al. on patients with ESRD and AF on dialysis, a standard dose of Apixaban 5 mg twice daily was associated with lower thromboembolic and mortality risks compared to Apixaban 2.5 mg twice a day or warfarin. Apixaban also demonstrated lower risks of major bleeding than warfarin [64]. Other studies comparing the efficacy of novel anticoagulants and warfarin failed to demonstrate consistent results, while, in a study, Rivaroxaban showed benefits in reducing cardiovascular events [78]. In other studies, Apixaban did not have any benefit over warfarin in reducing strokes [79,80]. Anticoagulation trials focusing on patients with end-stage renal disease (ESRD) face unique challenges due to the interplay of altered pharmacokinetics, increased thrombotic risks, and heightened bleeding tendencies. ESRD represents a higher-risk population with distinct clinical and physiological characteristics, necessitating careful considerations in trial design, anticoagulant selection, and outcome assessment (Table 3). Inconsistent and conflicting results from the studies have made it difficult to arrive at a consensus or devise guidelines for anticoagulant use [17]. Recent trials have tried to examine the use of anticoagulants in patients on hemodialysis (Table 4).

### 8.1. Patients with Afib and ESRD

ESRD increases the risk of Afib, thromboembolic events, and bleeding [82]. Recently, in 2023, a multicenter trial carried out on patients with Afib in ESRD showed no significant difference in bleeding and thrombosis between the Apixaban and the VKA groups [80]. In other studies, Rivaroxaban was superior to VKA in decreasing the cardiovascular risk with low bleeding potential in ESRD [78,83]. A 2020 study showed Apixaban has higher bleeding rates than no anticoagulation but no significant difference between the two groups in cardiovascular outcomes [66].

### 8.2. Patients with Stroke in ESRD

ESRD increases the risk of both hemorrhagic and ischemic stroke [84]. The risk is directly related to eGFR; with every 10 mL/min drop in eGFR, there is a 7% increase in the risk of stroke [25]. Also, the risk of stroke is seven times higher in the first year of dialysis compared to the general population [85]. Studies aimed at using antiplatelet agents for CVA prevention in patients with ESRD have yielded inconsistent results. A study showed promising results in decreasing the incidence of death and readmission for stroke. But other studies did not report a significant change in risk, even when associated with diabetes mellitus or severe hypertension [17].

## 9. Current Guidelines and the Need for Personalized Medicine

Patients with CKD/ESRD often present with various comorbidities, the most common of which are hypertension, diabetes, and cardiovascular diseases, such as coronary artery disease and congestive heart failure [86]. Cha & Han reported that around 70.8% of patients with ESRD have at least one or more comorbidities, which makes individualized therapy a necessity. Patient-centered medicine, also known as personalized medicine and precision medicine, among other names, is an updated approach to patient care that aims to tailor treatment based on the patient’s own needs and characteristics. This approach looks at the patient as a whole, not as a disease, by considering their medical record, genetics, socioeconomic level, lifestyle, expectations, and other factors that could potentially affect their response to treatment.

Especially in chronic conditions like CKD/ESRD, patient-centered medicine holds great potential for improving management outcomes as patients respond differently to treatment according to the individual’s demographic factors, genetic makeup, or the type of comorbidities they might have. For instance, African American populations are more likely to develop CKD/ESRD than those of other races, and have a higher prevalence of hypertension, which is one of the major causes of CKD. Therefore, health care providers may need to focus on early detection and management of hypertension in African American patients [87]. This further underscores the importance of a patient-centered approach in the health care of CKD/ESRD. Similarly, for particular patient populations, such as children, elderly, or pregnant patients, a personalized plan of treatment is vital. The coagulation profile in the pediatric population varies from that of adults as the blood-clotting system is not yet fully developed. Young children have lower levels of factors II, VII, IX, X, XI, XII and lower levels of von Willebrand factor, which is an important glycoprotein for platelet function and factor VIII stabilization. Children also have higher levels of fibrinogen, with low levels of plasminogen and tissue plasminogen activator, which are proteins involved in fibrinolysis. These differences in coagulation profile can affect the management of bleeding disorders in children. In the geriatric group, the risk of both bleeding and thrombosis is high [88,89]. The risk of thromboembolic events increases due to procoagulant changes associated with aging, and increases the risk of bleeding due to a decrease in platelet function and therapeutic interventions against thrombophilia. However, pregnant women have a higher risk of thrombosis due to the hypercoagulable state, with increased levels of clotting factors and fibrinogen in order to prevent excessive bleeding during delivery [90]. Since the coagulation profile varies among patient populations, the use of anticoagulants in patients with CKD/ESRD is complex and many international agencies recommend a patient-centered plan of management.

Furthermore, therapeutic drug monitoring (TDM) may be considered in advanced CKD patients to manage altered anticoagulant pharmacokinetics and reduce bleeding risks. UFH can be helped with aPTT monitoring for dose adjustment, while LMWH dosing benefits from an anti-Xa level assessment in advanced CKD. Warfarin therapy can be monitored with regular INR monitoring due to its narrow therapeutic window. Although routine TDM for DOACs is not yet standard, emerging assays may improve their safe use in renal impairment [91].

## 10. Future Directions

Future research on anticoagulation in ESRD should address several critical gaps identified in the recent studies. Future investigations should prioritize the cohort of patients with an eGFR < 30 mL/min, who remain markedly underrepresented in pivotal phase III DOAC trials. There is a critical need for well-powered, multicenter randomized controlled trials to rigorously compare the efficacy and safety profiles of DOACs versus warfarin within this population. In parallel, comprehensive pharmacokinetic and pharmacodynamic studies are warranted to elucidate optimal dosing paradigms in end-stage renal disease (ESRD), employing biomarker-guided therapeutic monitoring to enhance anticoagulant precision and mitigate hemorrhagic complications.

Furthermore, mechanistic studies dissecting the pathophysiological impact of uremia-induced alterations in platelet function and coagulation cascades are essential to inform the rational design of novel anticoagulants with favorable renal clearance and hemostatic profiles. Prospective, longitudinal cohort studies integrating advanced cardiovascular imaging modalities, biomarker assays, and wearable health technologies should be undertaken to characterize the long-term effects of anticoagulation on cardiovascular morbidity and mortality in ESRD patients with atrial fibrillation.

Finally, the development and validation of personalized anticoagulation strategies are imperative, incorporating variables such as residual renal function, dialysis modality, comorbidity burden, and pharmacogenomic determinants of drug metabolism and response. Investigations into combination pharmacotherapies and graduated dosing protocols may further refine therapeutic efficacy while minimizing adverse outcomes in this high-risk demographic.

## 11. Strengths and Limitations

This critical review offers a comprehensive and clinically relevant synthesis of the current evidence on anticoagulant use in CKD and ESRD, highlighting nuanced pharmacokinetic considerations, therapeutic dilemmas, and gaps in the existing guidelines. Its strength lies in the inclusion of diverse anticoagulant classes and a focus on individualized, patient-centered care, which is vital in a population with competing thrombotic and bleeding risks. However, this review is limited by the heterogeneity and scarcity of high-quality randomized controlled trials in this area, as well as potential publication bias. Despite these limitations, this study serves as an important reference to inform safer anticoagulation strategies in renal disease.

## 12. Conclusions

Anticoagulation in patients with CKD and ESRD presents a complex therapeutic challenge at the intersection of hematology, pharmacology, and internal medicine. The profound alterations in drug metabolism and excretion associated with impaired renal function significantly influence the pharmacokinetics and pharmacodynamics of anticoagulants, necessitating thoughtful dose adjustments and vigilant monitoring. Parenteral anticoagulants, such as heparin and LMWH, require dose adjustments in advanced CKD. UFH is preferred due to its shorter half-life, allowing for a rapid reduction in the anticoagulant effect, but needs cautious dose alteration based on aPTT. LMWH offers predictable pharmacokinetics and does not require monitoring, but its clearance is affected by renal function, requiring adjustments in CKD stages 4 and 5. While enoxaparin is commonly used, there is a paucity of data on the use of dalteparin and tinzaparin, warranting caution when dosing. Vitamin K antagonists, like warfarin, are commonly used but require caution in CKD due to their narrow therapeutic window, extensive drug interactions, and increased bleeding risk, and therefore dose adjustments based on estimated glomerular filtration rate (eGFR) are advised to maintain an international normalized ratio (INR) ≤ 4 and prevent hemorrhage. Even though warfarin is preferred in patients with ESRD, clear guidelines for warfarin use in dialysis patients are lacking, necessitating a careful risk–benefit assessment. Direct oral anticoagulants (DOACs) also require dose adjustments in CKD due to renal elimination. Apixaban is recommended for eGFR < 15 mL/min and dialysis patients, while data on other DOACs in CKD are limited. However, cautious use of DOACs is necessary in CKD, especially with eGFR < 30 mL/min, as phase 3 trials often exclude this population. Clear guidelines and standardized treatment for anticoagulation in CKD demand high-quality randomized controlled trials. Furthermore, owing to multiple comorbidities which influence the coagulation profile in CKD patients, a patient-centered approach to anticoagulation which considers the patient’s overall characteristics and response to treatment is required.

## Figures and Tables

**Figure 1 healthcare-13-01373-f001:**
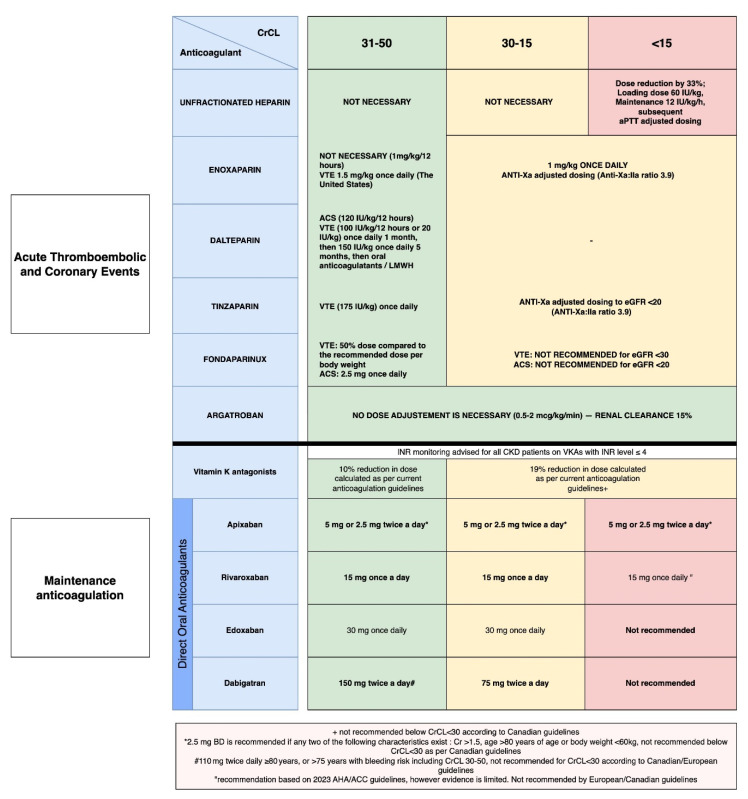
Evidence based algorithm for anticoagulation in patients with CKD [25,26,27,28,29,30,31,32,33,34,35,36,37,38,39,40,41,42,43,44,45,46,47,48,49,50].

**Table 1 healthcare-13-01373-t001:** KDIGO stages of CKD.

Stage	Description	GFR (mL/min)
Stage 1	Normal or high GFR	>90
Stage 2	Mild CKD	60–89
Stage 3A	Moderate CKD	45–59
Stage 3B	Moderate CKD	30–44
Stage 4	Severe CKD	15–29
Stage 5	End-Stage CKD	<15

**Table 2 healthcare-13-01373-t002:** Comparative effectiveness and safety of antithrombotic therapy in patients with atrial fibrillation and end-stage renal disease (ESRD).

Year	Author	Design	Population	Stroke/Thromboembolism	Bleeding
1997–2008	Olesen et al. [60]	Retrospective observational cohort Warfarin use	Patients discharged with non-valvular AF	Warfarin use HR, 0.44; 95% CI, 0.26–0.74; *p* = 0.002	HR, 1.27; 95% CI, 0.911–1.77; *p* = 0.15
1998–2007	Shah et al. [61]	Retrospective cohort Warfarin use	Patients aged ≥ 65 years ad-mitted to a hospital with a diagnosis of AF	Stroke: HR, 1.14; 95% CI, 0.78–1.67	Bleeding: HR, 1.44; 95% CI, 1.13–1.85
2006–2013	Waddy et al. [62]	Retrospective observational cohort Warfarin use	patients with ESRD who initiated HD and subsequently diagnosed with AF	Ischemic stroke: aHR, 0.79; 95% CI, 0.66–0.95	GI bleeding: HR, 0.97; 95% CI, 0.77–1.2 Hemorrhagic stroke: HR, 1.2; 95% CI, 0.6–2.2
2009–2013	Yoon et al. [63]	Propensity matched cohort Warfarin use	Patients with ESRD and AF	Ischemic stroke: HR, 0.95; 95% CI, 0.78–1.15; *p* = 0.569	GI bleeding with the warfa- rin use (7.5%) and no warfarin use (6.6%), *p* = 0.208 Hemorrhagic stroke: HR, 1.56, 95% CI, 1.10–2.22; *p* = 0.013
2010–2015	Siontis et al. [64]	Retrospective cohort DOACs	Patients with AF and ESRD undergoing dialysis	Stroke/systemic embolism for Apixaban vs. warfarin: HR, 0.88; 95% CI, 0.69–1.12; *p* = 0.29	GI bleeding for Apixaban vs. warfarin: HR, 0.86; 95% CI, 0.72–1.02; *p* = 0.09 Intracranial bleed for Apixaban vs. warfarin: HR, 0.79; 95% CI 0.49–1.26; *p* = 0.32
2019	Kuno et al. [65]	Meta-analysis DOACs	Studies that investigated the efficacy and safety of different OAC strategies in patients with AF on long-term dialysis	Apixaban 2.5 mg: HR, 1.00; 95% CI, 0.52–1.93 Apixaban 5 mg: HR, 0.59; 95% CI, 0.30–1.17	Apixaban 2.5 mg: HR, 1.40; 95% CI, 1.07–1.82 Apixaban 5 mg: HR 1.41; 95% CI, 1.07–1.88
2020	Mavrakanas et al. [66]	Propensity cohort DOACs	Patients on maintenance dialysis with incident, non-valvular AF treated with Apixaban	Ischemic stroke for Apixaban as compared to no treatment: HR, 0.85; 95% CI, 0.36–1.98; *p* = 0.71	Major bleeding for Apixaban as compared to no treatment: HR, 2.76; 95% CI, 1.38–5.52; *p* = 0.004 Hemorrhagic stroke for Apixaban as compared to no treatment: HR, 1.89; 95% CI, 0.65–5.49; *p* = 0.24

**Table 3 healthcare-13-01373-t003:** Challenges in conducting anticoagulation trials in ESRD.

Challenge	Details
Altered Pharmacokinetics and Dynamics	Anticoagulants show variable metabolism and excretion in ESRD. DOACs like Rivaroxaban and dabigatran accumulate, increasing bleeding risk, while warfarin’s effects vary due to altered protein binding and vitamin K fluctuations.
Thrombotic and Bleeding Risk Paradox	Patients with ESRD exhibit both hypercoagulability and increased bleeding risk due to platelet dysfunction, vascular endothelial abnormalities, and altered coagulation factors. Balancing these risks in trials is complex.
Study Population Diversity	Patients with ESRD often have comorbidities (e.g., diabetes, cardiovascular disease), which may confound trial results. Hemodialysis also influences anticoagulant pharmacokinetics and adds vascular complications.
Endpoints and Outcome Assessment	Trials must address both thrombotic events (e.g., DVT, PE, arteriovenous fistula thrombosis) and bleeding risks. Mortality and quality-of-life metrics are critical for this population.

**Table 4 healthcare-13-01373-t004:** Randomized trials on the use of anticoagulants in patients on hemodialysis.

**Study and Year**	**Type of Study**	**Objectives**	**Result or Conclusions**
Wang et al. 2015 [81]	Open-label, parallel-group, single-dose study	Pharmacokinetics of a single dose of 5 mg Apixaban in patients with ESRD on hemodialysis compared to normal subjects	Hemodialysis had little impact on Apixaban clearance. ESRD led to a modest increase in the AUC of Apixaban but no increase in Cmax.
De Vriese et al. 2021 [78]	Multicenter RCT	Safety and efficacy of VKA vs. Rivaroxaban in hemodialysis patients with Afib	Compared to VKA: Rivaroxaban significantly reduced fatal and nonfatal cardiovascular events. The Rivaroxaban group had reduced major bleeding complications.
RENAL-AF trail 2022 [79]	Multicenter RCT	5 mg twice-daily Apixaban daily versus dose-adjusted warfarin in patients with Afib on hemodialysis	Apixaban failed to demonstrate any significant reduction in stroke cases, but there were increased bleeding rates.
AXADIA-AFNET 8 study 2022 [80]	RCT	Compare Apixaban with Vitamin K antagonist in patients on chronic hemodialysis	No differences in safety or efficacy outcomes between the two drugs.

## Data Availability

Not applicable.

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
