# Peer review of "Anticoagulation in Patients with End-Stage Renal Disease: A Critical Review"

_healthcare, 2025, doi:10.3390/healthcare13121373_

Round 1
Reviewer 1 Report
Comments and Suggestions for Authors
I read with interest the paper titled "Anticoagulation in Patients with ESRD: A critical review"
The paper is well written and I have only minor comments to add:
1. Affiliation should be added to each of the authors
2. Abstract should have introduction, rather than start with the objective.
3. References should be confirmed (e.g.: ((Cozzolino et al. 2018),([CSL STYLE 63
ERROR: reference with no printed form.])) - line 63) - in addition, they should be in line with journal requirements.
4. Authors state that "search terms included combinations of...", however, when tried to replicate the search, no results were found that are coincident with what authors state that was made. The study is not replicable in this way, I suggest to clarify the keywords used.
5. Strengths and limitations of the article should be discussed in a separate section.
Author Response
Thank you for your kind and encouraging review. We truly appreciate the time you took to engage with our work so thoughtfully. Your positive comments affirmed the strengths of our approach, and your constructive feedback provided meaningful insights that will help us further refine and improve our research. It’s incredibly rewarding to know that our work resonated with you.
Comment 1: Affiliation should be added to each of the authors.
Authors’ response: Thank you for pointing it out. Affiliations for all authors have now been included in the manuscript as per the journal’s submission requirements.
Comment 2: Abstract should have an introduction, rather than start with the objective.
Authors’ response: Thank you for pointing it out. We have revised the abstract to include a brief introductory context before stating the objective.
Comment 3: References should be confirmed and aligned with journal requirements.
Authors’ response: Thank you for pointing it out. All references have been carefully checked and corrected for accuracy and formatting according to the MDPI Healthcare referencing style. The previous CSL STYLE errors have been resolved.
Comment 4: Clarify keywords used in the search strategy.
Authors’ response: Thank you for pointing it out. The search strategy has been revised for transparency. The specific keywords and Boolean operators used are now clearly listed in the methodology section.
Comment 5: Strengths and limitations of the article should be discussed in a separate section.
Authors’ response: Thank you for pointing it out. A separate “Strengths and Limitations” section has been added prior to the conclusion to provide a balanced perspective on the review.
Reviewer 2 Report
Comments and Suggestions for Authors
The manuscript will definitely add to the body knowledge especially in the field of medicine and pharmacology. However, it was not well structured as well as the research gap and objectives of the review was not clearly define. This query the scientific merit of the review article. In addition to these, there are many typo and technical errors in the write. Likewise, the conclusion does not align with their finding.
The manuscript require major revision before consider for publication. The authors are require to address this following comments to improve the quality of work.
1. The title needs to align with the contents of the review. In addition, the authors need to write on full before subsequent abbreviation (CCk).
2. The authors need to only include literatures relevant to the subject matters for the review.
3. The authors should clearly define the rationale of study or problem state and clearly define the objectives of the study.
4. The authors should limit the literatures provided basing it on the scope of the literature.
5. The authors need to restructure the review article following the instruction provided by the Journal's authors guideline.
6. The figures and tables are require to the format in line the aithors guildline.
7. The authors needs to avoid using abbreviation without subsequently write in detail.
8. The search methods scope need to be review by the authors.
9. The conclusion statements need to be reframe to align with the content.
10. The authors need to provide information of the authors contribution.
11. The references needs to be review by the authors. There are alot of punctuation errors and lack uniform referencing style.
12. The manuscript is too lengthen, the authors need to check the authors guildline.
Comments on the Quality of English LanguageThe manuscript require language editing.
Author Response
Thank you for your thoughtful review and valuable feedback — we're grateful you appreciated our work and took the time to provide such encouraging and insightful comments.
Comment 1: The title needs to align with the content; abbreviations should be written in full on first use.
Authors’ response: The title has been revised for better alignment with the content. All abbreviations, including “ESRD”, are now fully written out upon first mention.
Comment 2: Only include literature relevant to the subject.
Authors’ response: Thank you for pointing it out. We have reviewed and excluded references not directly related to anticoagulation in ESRD to improve focus and relevance.
Comment 3: Define the rationale and objectives clearly.
Authors’ response: Thank you for the comment. The introduction section has been expanded to clearly state the rationale and objectives of the review.
Comment 4: Limit literature to the scope of the review.
Authors’ response: Thank you for the comment. The literature scope has been refined, and unrelated content has been removed to maintain thematic consistency.
Comment 5: Restructure the article according to the journal’s format.
Authors’ response: The manuscript has been restructured in accordance with MDPI Healthcare’s guidelines for review articles.
Comment 6: Figures and tables should follow author guidelines.
Authors’ response: All figures and tables have been reformatted and aligned with the journal’s formatting standards.
Comment 7: Avoid unexplained abbreviations.
Authors’ response: Thank you for the comment. Abbreviations are now consistently defined on first use throughout the manuscript.
Comment 8: Review the scope of the search methods.
Authors’ response: The search methodology section has been expanded to provide a detailed description, including databases, timeframe, inclusion/exclusion criteria, and study selection process.
Comment 9: Reframe the conclusion to align with findings.
Authors’ response: The conclusion section has been rewritten to reflect and summarize the findings accurately, with practical clinical implications.
Comment 10: Provide author contributions.
Authors’ response: The “Author Contributions” section has been added according to the journal template.
Comment 11: Review reference formatting.
Authors’ response: All references have been reviewed, corrected for punctuation, and standardized to MDPI formatting.
Comment 12: The manuscript is too lengthy.
Authors’ response: The manuscript has been revised for conciseness by removing redundancies and streamlining the content in several sections.
Reviewer 3 Report
Comments and Suggestions for Authors
The manuscript entitled Anticoagulation in Patients with ESRD: A critical review is work characterised by high quality, novelty of topic and seems to be interesting for Healthcare journal readers.
The graphical background is clearly presented and informative. The discussion and conclusions are appropriate. The number of references is sufficient.
I have two suggestions only,
- The authors should introduce the paragraph with best practices and suggestions for clinicians according to anticoagulation in ESRD.
- Please define, if therapeutic drug monitoring (TDM) is suggested for anticoagulants during ESRD.
- Please redact manuscript and references according to MDPI/Journal guidelines.
To sum up, the manuscript is recommend to publication after introducing suggestions. Good work - congratulations.
Author Response
We are extremely glad that you liked the work. Many thanks for your comments and detailed feedback. The comments have been addressed as follows:
Comment 1: Introduce best practices and suggestions for clinicians.
Authors’ response: Thank you. We have revised the paragraph outlining best practices and clinician recommendations based on current evidence in the discussion section.
Comment 2: Define if therapeutic drug monitoring (TDM) is suggested.
Authors’ response: Thank you for the valuable suggestion. We have added a specific discussion on the role of TDM for anticoagulants in ESRD, including scenarios where it may be applicable.
Comment 3: Redact manuscript and references according to MDPI guidelines.
Authors’ response: The manuscript and references have been revised and formatted strictly according to MDPI Healthcare guidelines.
Reviewer 4 Report
Comments and Suggestions for Authors
Comments to the authors:
- The manuscript contains numerous grammatical errors and awkward sentence constructions; a thorough language edit by a professional or native English speaker is strongly recommended.
- Several sections of the manuscript are repetitive, especially discussions on bleeding vs. thrombotic risk and warfarin pharmacokinetics; please condense the content to avoid redundancy and improve focus.
- The referencing format is inconsistent throughout the manuscript. Some citations are incomplete or incorrectly formatted (e.g., “CSL STYLE ERROR”); please revise and ensure all references follow the journal’s guidelines.
- The methodology section describing the review process lacks clarity regarding inclusion/exclusion criteria, time frame, and study quality assessment; consider elaborating to enhance transparency.
- While the review is comprehensive, the analysis remains largely descriptive. Aim to include more critical evaluation of the evidence, especially when comparing different anticoagulants.
- Some figures and tables (e.g., Figure 2 and Table 4) are not clearly integrated into the main text or are insufficiently described; improve captions and contextual referencing.
- The discussion on management protocols would benefit from a concise summary table comparing all anticoagulant options (UFH, LMWH, warfarin, DOACs) across CKD/ESRD stages.
- The proposed algorithm for anticoagulation is useful but should be more clearly formatted and supported by referenced clinical evidence.
- The manuscript would benefit from a clearer conclusion section summarizing practical clinical recommendations and take-home messages for physicians.
- A dedicated section on personalized medicine is included but would be strengthened by integrating specific clinical examples and linking more directly to anticoagulation decisions.
- The future directions section should outline more specific research gaps and propose targeted questions for upcoming trials, especially regarding DOAC use in ESRD populations.
- Please ensure consistent terminology throughout the manuscript (e.g., ESRD vs. CKD stage 5) and avoid mixing terms without clarification.
Comments on the Quality of English Language
nil
Author Response
We would like to express our sincere gratitude to you and the reviewers for your constructive and insightful comments on our manuscript. We have carefully addressed all suggestions and revised the manuscript accordingly. Please find below our detailed point-by-point responses to each reviewer’s comments. All changes have been highlighted in the revised version of the manuscript and formatted according to the MDPI Healthcare journal guidelines.
Comment 1: Numerous grammatical errors; language editing needed.
Authors’ response: Thank you for the valuable suggestion. The manuscript has undergone comprehensive language editing by a native English speaker to improve clarity, grammar, and flow.
Comment 2: Redundant content in sections on bleeding vs thrombotic risk and warfarin pharmacokinetics.
Authors’ response: Thank you for the valuable suggestion. These sections have been condensed to eliminate redundancy and enhance clarity.
Comment 3: Inconsistent referencing format.
Authors’ response: Reference formatting has been corrected throughout the manuscript in line with MDPI’s standards.
Comment 4: Lack of clarity in review methodology.
Authors’ response: We have expanded the methodology section to clarify inclusion/exclusion criteria, timeframe, databases used, and quality assessment of included studies.
Comment 5: Descriptive rather than critical analysis.
Authors’ response: We have revised the discussion to include more critical appraisal of the evidence, especially in comparing different anticoagulant options.
Comment 6: Figures/tables not integrated or described well.
Authors’ response: All figures and tables have been revised for clarity, with improved captions and integration into the text.
Comment 7: Improve discussion of management protocols via summary table.
Authors’ response: Thank you for pointing out the need. Figure 2 has been made with the same intention. It not only provides an algorithm but gives a summary of management protocols.
Comment 8: Format proposed algorithm better and support it with references.
Authors’ response: The proposed algorithm has been reformatted for clarity and now includes supporting references from relevant guidelines and studies.
Comment 9: Clearer clinical conclusions needed.
Authors’ response: A revised conclusion section now provides practical recommendations and take-home messages for clinicians.
Comment 10: Strengthen personalized medicine section with clinical examples.
Authors’ response: We have enriched this section with specific examples and linked them more directly to decision-making in anticoagulation.
Comment 11: Add more specific research gaps in future directions.
Authors’ response: The “Future Directions” section now outlines targeted research questions, especially on DOAC use in ESRD.
Comment 12: Ensure consistent terminology.
Authors’ response: We have reviewed and standardized terminology throughout (e.g., consistently using “ESRD” or “CKD stage 5” with clarifications where needed).
Round 2
Reviewer 4 Report
Comments and Suggestions for Authors
all comments are address well.